

# Fluconazole worsened lung inflammation, partly through lung microbiome dysbiosis in mice with ovalbumin-induced asthma

Jesadakorn Worasilchai[1,2], Piyapat Thongchaichayakon[1,2], Kittipat Chansri[1,2], Supichaya Leelahavanichkul[1,3], Vathin Chiewvit[1], Peerapat Visitchanakun[2], Poorichaya Somparn[3] and Pratsanee Hiengrach[4,5]

[1] Center of Excellence on Translational Research in Inflammation and Immunology (CETRII), Department of Microbiology, Faculty of Medicine, Chulalongkorn University, Bangkok, Thailand
[2] Department of Microbiology, Faculty of Medicine, Chulalongkorn University, Bangkok, Thailand
[3] Center of Excellence in Systems Biology, Research Affairs, Faculty of Medicine, Chulalongkorn University, Bangkok, Thailand
[4] Department of Microbiology, Faculty of Medicine, Khon Kaen University, Khon Kaen, Thailand
[5] Research and Diagnostic Center for Emerging Infectious Diseases (RCEID), Faculty of Medicine, Khon Kaen University, Khon Kaen, Thailand

Corresponding author
Pratsanee Hiengrach,
pratshi@kku.ac.th

## ABSTRACT

Innate immunity in asthma may be influenced by alterations in lung microbiota, potentially affecting disease severity. This study investigates the differences in lung inflammation and microbiome between asthma-ovalbumin (OVA) administered with and without fluconazole treatment in C57BL/6 mice. Additionally, the role of inflammation was examined in an *in vitro* study using a pulmonary cell line. At 30 days post-OVA administration, allergic asthma mice exhibited increased levels of IgE and IL-4 in serum and lung tissue, higher pathological scores, and elevated eosinophils in bronchoalveolar lavage fluid (BALF) compared to control mice. Asthma inflammation was characterized by elevated serum IL-6, increased lung cytokines (TNF-α, IL-6, IL-10), and higher fungal abundance confirmed by polymerase chain reaction (PCR). Fluconazole-treated asthma mice displayed higher levels of cytokines in serum and lung tissue (TNF-α and IL-6), increased pathological scores, and a higher number of mononuclear cells in BALF, with undetectable fungal levels compared to untreated mice. Lung microbiome analysis revealed similarities between control and asthma mice; however, fluconazole-treated asthma mice exhibited higher Bacteroidota levels, lower Firmicutes, and reduced bacterial abundance. Pro-inflammatory cytokine production was increased in supernatants of the pulmonary cell line (NCI-H292) after co-stimulation with LPS and beta-glucan (BG) compared to LPS alone. Fluconazole treatment in OVA-induced asthma mice exacerbated inflammation, partially due to fungi and Gram-negative bacteria, as demonstrated by LPS+BG-activated pulmonary cells. Therefore, fluconazole should be reserved for treating fungal asthma rather than asthma caused by other etiologies.

Subjects Microbiology, Allergy and Clinical Immunology, Respiratory Medicine
Keywords Ovalbumin, Asthma, Microbiome, Fungi

## INTRODUCTION

Asthma is a chronic inflammatory condition that affects airways, leading to recurrent respiratory issues manifestations, including wheezing, shortness of breath, chest tightness, and cough (*Nguyen & Nasir, 2024*). Asthma is commonly divided into two types based on phenotype, including type-2 (T2) high asthma and type-2 (T2) low asthma. As such, T2-high asthma is triggered by allergens (house dust mites, animal dander, and pollen) and environmental factors (smoke, pollution, cold air, and microorganisms) (*Pechsrichuang & Jacquet, 2020*). The airway inflammation in T2-high asthma is closely associated with cytokines from type 2 T helper cells (Th2), particularly interleukin-4 (IL-4), which leads to eosinophilia, excessive mucus production, bronchial hyperresponsiveness, and immunoglobulin E (IgE) production (*León & Ballesteros-Tato, 2021*). Notably, this Th2-mediated response is considered the main underlying mechanism of asthma (*Jacquet, 2013*). On the other hand, T2-low asthma involves allergen-stimulated antigen-presenting cells that not only initiate Th2 responses but, in some cases, activate other types of immune cells, for example, Th17-mediated neutrophils and/or Th1-associated macrophages (*Finn & Bigby, 2009*). These immune cells secrete several important cytokines in asthma (IL-8, IL-17, and IL-22) that might be associated with the impacts of pollutants and/or microorganisms in the asthmatic lung (*Kyriakopoulos et al., 2021*).

In mice, ovalbumin (OVA) combined with an immunostimulatory adjuvant, such as aluminum hydroxide (alum), serves as a well-established model for asthma, characterized by several features, including mucus production, airway reactivity, and inflammatory cell infiltration. This model is widely utilized in various research topics related to asthma (*Bhunyakarnjanarat et al., 2024*; *Casaro et al., 2019*). Equally, intraperitoneal sensitization followed by intratracheal OVA challenge (a 45 kDa glycoprotein in egg whites) effectively activates IgE-associated Th2, increasing IL-4 production (*Kim, Song & Lee, 2019*). Accordingly, normal microbiota in the lung comprises several bacteria, such as *Prevotella*, *Streptococcus*, and *Veillonella*, and an imbalance of microbiota, referred to as "dysbiosis" is mentioned in several lung diseases (*Li, Li & Zhou, 2024*) and some patterns of lung microbiome linked to the severity of asthma (*Zheng et al., 2021*). Indeed, the innate phagocytic cells (neutrophils and macrophages) demonstrate an important role in the balance of normal microbiota, and the interference of this balance, for example, an increase or decrease in these cells, might alter normal regulation, causing dysbiosis (*Hiengrach et al., 2022*). Not only bacteria but fungi (*Candida albicans* in humans) are also the second most abundance organisms in the oropharyngeal, upper respiratory, and gastrointestinal tracts (*Amornphimoltham et al., 2019*), which might occasionally pass from the upper respiratory tract into the lung (lower respiratory tract), especially during the asthmatic attack (increased mucus and labored respiration) (*van Tilburg Bernardes, Gutierrez & Arrieta, 2020*). Also, the alteration of the bacterial population by fungi and *vice versa* is well-known, especially from the gut microbiota analysis, partly through symbiosis and antagonistic communication (*Santus, Devlin & Behnsen, 2021*).

Additionally, fungi-associated asthma, referred to as "fungal sensitization" or "severe asthma with fungal sensitization (SAFS)," is clinically severe asthma with a high immune

response against fungal allergens (antigens), which is one of the important causes of severe asthma in patients. These allergens, particularly from *Aspergillus* spp., *Penicillium* spp., *Alternaria* spp., and *Cladosporium* spp., significantly exacerbate respiratory symptoms (*Agarwal, Muthu & Sehgal, 2023*). Interestingly, fluconazole, a commonly used antifungal drug, has been reported to attenuate the severity of late-onset asthma, possibly by reducing fungal burdens (*Ward et al., 1999*). Conversely, fluconazole has been observed to decrease *Candida* spp. at various body sites, alter bacterial populations, and increase asthma severity (*van Tilburg Bernardes, Gutierrez & Arrieta, 2020*). Hence, the effects of fluconazole on asthma remain inconclusive. Then, we hypothesized that immune responses during active asthma might elevate fungal abundance in the lower respiratory tract, and fluconazole might be beneficial during the asthmatic attack. Due to the limited data on fungal exploration in OVA mice and the inconclusive findings regarding fluconazole treatment for asthma, we investigated the effects of fluconazole on asthmatic mice together with lung microbiome analysis and the *in vitro* experiments on the impact of bacteria and fungi in NCI-H292 cell line (mucoepidermoid carcinoma cell line).

## MATERIALS AND METHODS

### Animal and animal model

The animal study protocol received approval from the Institutional Animal Care and Use Committee of the Faculty of Medicine, Chulalongkorn University, under NIH guidelines (SST 016/2566). To avoid the influence of estrogen on asthma, male C57BL/6 8-week-old mice (20–25 g) were obtained from Nomura Siam International, Pathumwan, Bangkok, Thailand. Briefly, all mice were housed in a specific pathogen-free (SPF) animal research facility with a controlled environment, including a temperature of 24 ± 2 °C, 50% relative humidity, and a 12-h light-dark cycle, with light provided from 7:00 a.m. to 7:00 p.m. Six mice were housed in cages with corncob bedding, which was autoclaved before use. The mice had continuous access to a standard sterile diet and autoclaved tap water, provided *via* a feeder and a permanently placed bottle within the cage. All mice were randomly divided into four groups, each containing six mice, which included: (i) the normal control (Control), (ii) OVA-induced asthma (Asthma), (iii) Fluconazole administration (Antifungal control), and (iv) the Fluconazole-treated OVA-induced asthma group. Six mice in each group were randomly divided into two subgroups of three mice each, and the experiments were independently repeated twice. The OVA-induced asthma mice were given a combination of 50 µL of Ovalbumin (Sigma-Aldrich, St. Louis, MA, USA) and 1.6 mg of Alum Adjuvant (aluminum hydroxide; Thermo Scientific, Rockford, IL, USA) through intraperitoneal injection (i.p.) on day 0 and day 7 before the sensitization by intratracheal (i.t.) administration of a 50 µg OVA solution on days 14, 21, and 22. To investigate the antifungal treatment, a daily oral administration of 250 µL of a 2 mg/mL fluconazole solution was performed for 1 week as a previous publication (*Heng, Jiang & Chu, 2021*). This was conducted in groups receiving fluconazole alone and asthma with fluconazole. Twenty-four hours after the final dose of fluconazole, all mice were humanely euthanized *via* cardiac puncture using an overdose of isoflurane (Abbott Laboratories Ltd.,

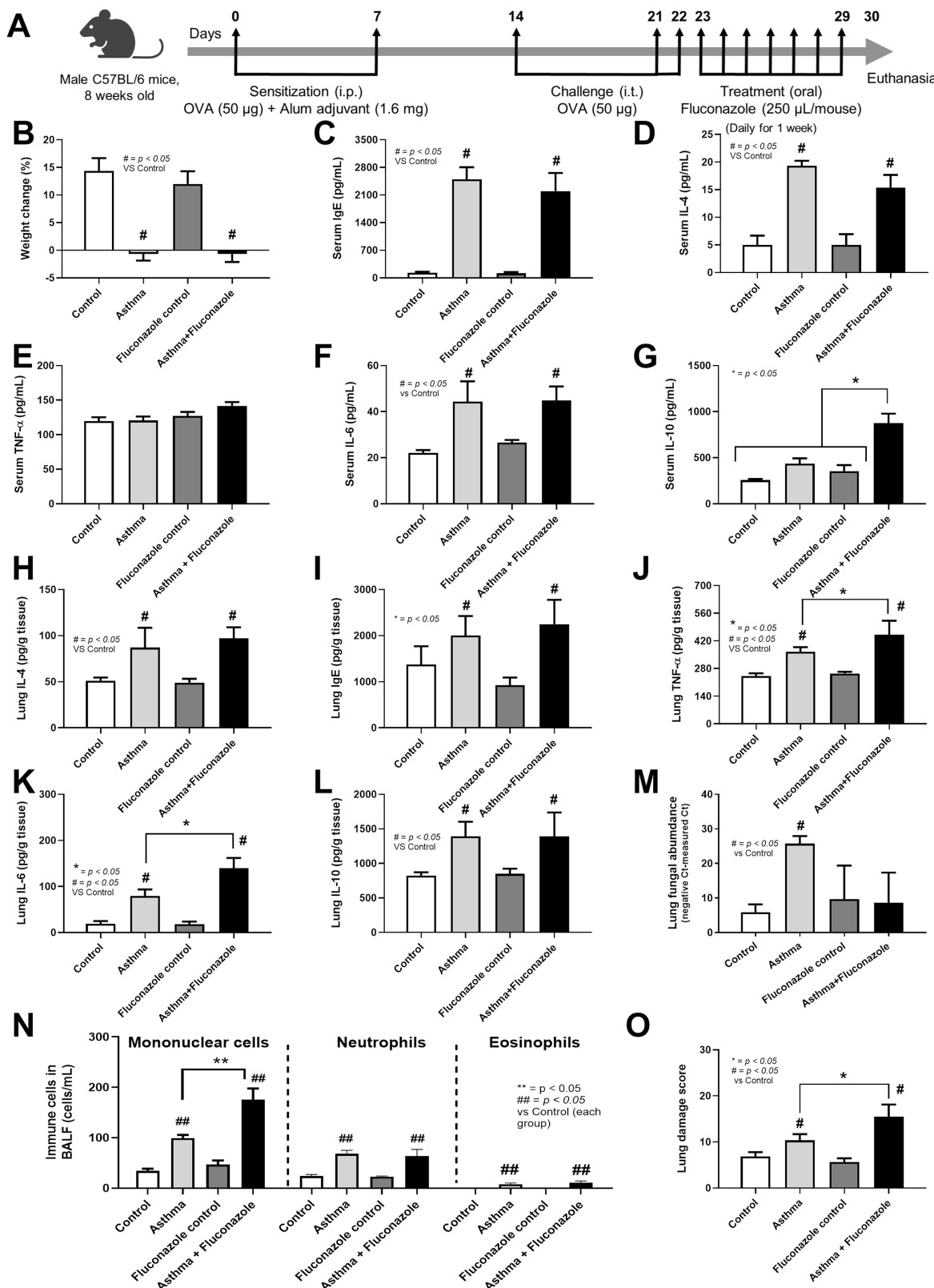

**Figure 1** Schema of the establishment of allergic asthma model (see Materials and Methods) (A) and characteristics of control, asthma, fluconazole in normal mice, and fluconazole in asthma as indicated by body weight change (B), serum Ig E (C), and serum IL-4 (D), serum inflammatory cytokines (TNF-$\alpha$, IL-6, and IL-10) (E–G), lung homogenates (IL-4, Ig E, TNF-$\alpha$, IL-6, IL-10, and fungal abundance)

# PeerJ

**Figure 1 (continued)**
**(H–M), immune cells in bronchoalveolar lavage fluid (BALF) (N), and lung damage score (O) are demonstrated, (*n* = 6/group).** The data are shown as the mean ± SE, #, ##; *p* < 0.05 *vs.* Control, and *, **; *p* < 0.05 *vs.* Asthma + Fluconazole using ANOVA with Tukey's analysis. Portions of this figure created with BioRender.com.               

Bangkok, Thailand). Bronchoalveolar lavage fluid (BALF) and lung tissue were collected for analysis according to the experimental schema (Fig. 1A).

## Mouse sample analysis

Serum cytokines (IL-4, TNF-α, IL-6, and IL-10) and IgE were measured by the enzyme-linked immunosorbent assay (ELISA) (Invitrogen, Waltham, MA, USA). Additionally, lungs (0.1 g) were homogenized through sonication (20 s of pulse-on, 5 s of pulse-off on ice for a total of 45 s) before centrifugation and evaluated tissue cytokines (IL-4, TNF-α, IL-6, and IL-10) and IgE using ELISA (Invitrogen). Fungal identification in lung samples was performed *via* culture and PCR, following established protocols (*Khot & Fredricks, 2009*). Briefly, mouse lung tissue was homogenized by a lysis buffer at 65 °C for 3 h, digested (mechanical bead beater for 20 min at 15 vibrations per second) before the extraction of metagenomic DNA by Phenol: Chloroform method, and amplified by the universal Internal Transcribed Spacer (ITS) eukaryotic primers; ITS3 (forward: 5′-GCATCGATGAAGAACGCAGC-3′) and ITS4 (reverse: 5′-TCCTCCGCTTATTGATATGC-3′). Furthermore, the lung microbiome analysis was performed according to the previous protocol (*Toju et al., 2012*). Briefly, lung tissue (0.1 g) was used for purifying metagenomic DNA as a template for amplifying the V3-V4 region of the 16S rRNA-encoding gene. It was sequenced by the Illumina Miseq sequencing platform (Illumina, San Diego, CA, USA). Universal prokaryotic 16S rRNA primers, 515F (forward: 5′-GTGCCAGCMGCCGCGGTAA-3′) and 806R (reverse: 5′-GGACTACHVGGGTWTCTAAT-3′), supplemented with Illumina adapter and Golay barcode sequences, were employed to construct the bacteriome sequencing library for the 16S rRNA gene. The detected mouse lung 16S rRNA sequences were deposited in the NCBI database (accession number PRJNA1019875).

The histology, the lung tissue was preserved in a 10% paraformaldehyde solution, followed by embedding in paraffin and sectioning at a 5 μM thickness before staining by Hematoxylin and eosin (H&E) or Masson's Trichrome colors for pulmonary morphology and lung fibrosis, respectively, following standard protocol (*Udompornpitak et al., 2023*). The histological analysis was conducted by two blinded analysts in 10 randomly selected areas per slide based on immune cell infiltration, alveolar thickening, and parabronchial infiltration with the following scores: 0 (indicating no discernible findings) to 4 (indicative of diffuse inflammation with over 50 visualized lumens), as previously published (*Phuengmaung et al., 2022*).

For the collection of BALF, mice were euthanized with an overdose of isoflurane following a previously published protocol (*Kalidhindi, Ambhore & Sathish, 2021*). In short, a small horizontal incision was made in the trachea using a sharp scalpel, then 1× phosphate-buffered saline (PBS), pH 7.4 (1 mL), was slowly injected into the cannula and the injected PBS back into the syringe, and repeated the process with an additional 1 mL of

PBS. The BALF was centrifuged at 2,000 × *g* for 5 min at 4 °C and the cell pellets were resuspended in 1 mL of PBS and were stained with Wright-Giemsa color (Sigma-Aldrich) using a standard protocol (*Tongthong et al., 2023*). The immune cells (mononuclear cells, eosinophils, and neutrophils) were subsequently analyzed by two blinded pathologists in 10 randomly selected fields (×400 magnification) per slide.

### The *in vitro* experiments

To explore the impacts of fungal and bacterial components on the human pulmonary mucoepidermoid carcinoma cell line (H292) (ATCC, Manassas, VA, USA) were cultured in Dulbecco's Modified Eagle Medium (DMEM), supplemented with 10% heat-inactivated fetal bovine serum (FBS) from Gibco, Carlsbad, CA, USA, and 1% PenStrep. The cell culture was maintained at 37 °C in a 5% $CO_2$ incubator. Subsequently, H292 cells at a concentration of $1 \times 10^6$ cells per well were incubated with lipopolysaccharides (LPS) (*Escherichia coli* 026: B6; Sigma-Aldrich, St. Louis, MA, USA) (a representative of Gram-negative bacterial cell wall) (100 ng/mL) with or without $(1\rightarrow 3)$-β-D-glucan (BG), a representative fungal cell wall component, at concentrations of 0.1 or 1 mg/mL at 37 °C in a 5% $CO_2$ environment. Cytokines (TNF-α, IL-6, IL-8, IL-10) in supernatant samples were measured using ELISA (Invitrogen).

### Statistical analysis

Data were presented as mean ± standard error (SE) using SPSS 11.5 and GraphPad Prism version 7.0 for the statistical analysis. One-way analysis of variance (ANOVA) followed by Tukey's analysis and Student's t-test was used for multiple groups and two group comparisons, respectively. A *p-value* of less than 0.05 was considered statistically significant.

## RESULTS

### Fluconazole worsened lung inflammation in OVA-induced asthmatic mice

The experimental setup for *in vivo* test is illustrated in Fig. 1A. Asthmatic mice with and without fluconazole exhibited retardation of weight gain (Fig. 1B), increased IgE and IL-4 in serum (Figs. 1C and 1D), increased serum IL-6 but not serum TNF-α (increased serum IL-10 only in asthma with fluconazole) (Figs. 1E–1G), and prominent IL-4 and IgE in the lung homogenates (Figs. 1H and 1I). For the local lung inflammation, lung TNF-α and IL-6, but not IL-10, in the OVA-fluconazole group was higher than the OVA mice (Figs. 1J–1L), while OVA mice demonstrated increased fungal burden in the lungs, as indicated by elevated ITS gene expression, a marker of fungal DNA (Fig. 1M). Histological analysis revealed greater lung damage score and prominent monocytes in bronchoalveolar lavage fluid (BALF), but not neutrophil nor eosinophil, in fluconazole-treated asthmatic mice compared with the OVA asthma group (Figs. 1N–1O and Fig. 2). Notably, the more prominent mononuclear cell infiltration and subepithelial fibrosis in fluconazole-treated asthma over asthma alone was indicated in the Hematoxylin and Eosin (Fig. 2 upper) and Masson's Trichrome (Fig. 2 lower) colors, respectively. However, fluconazole-induced

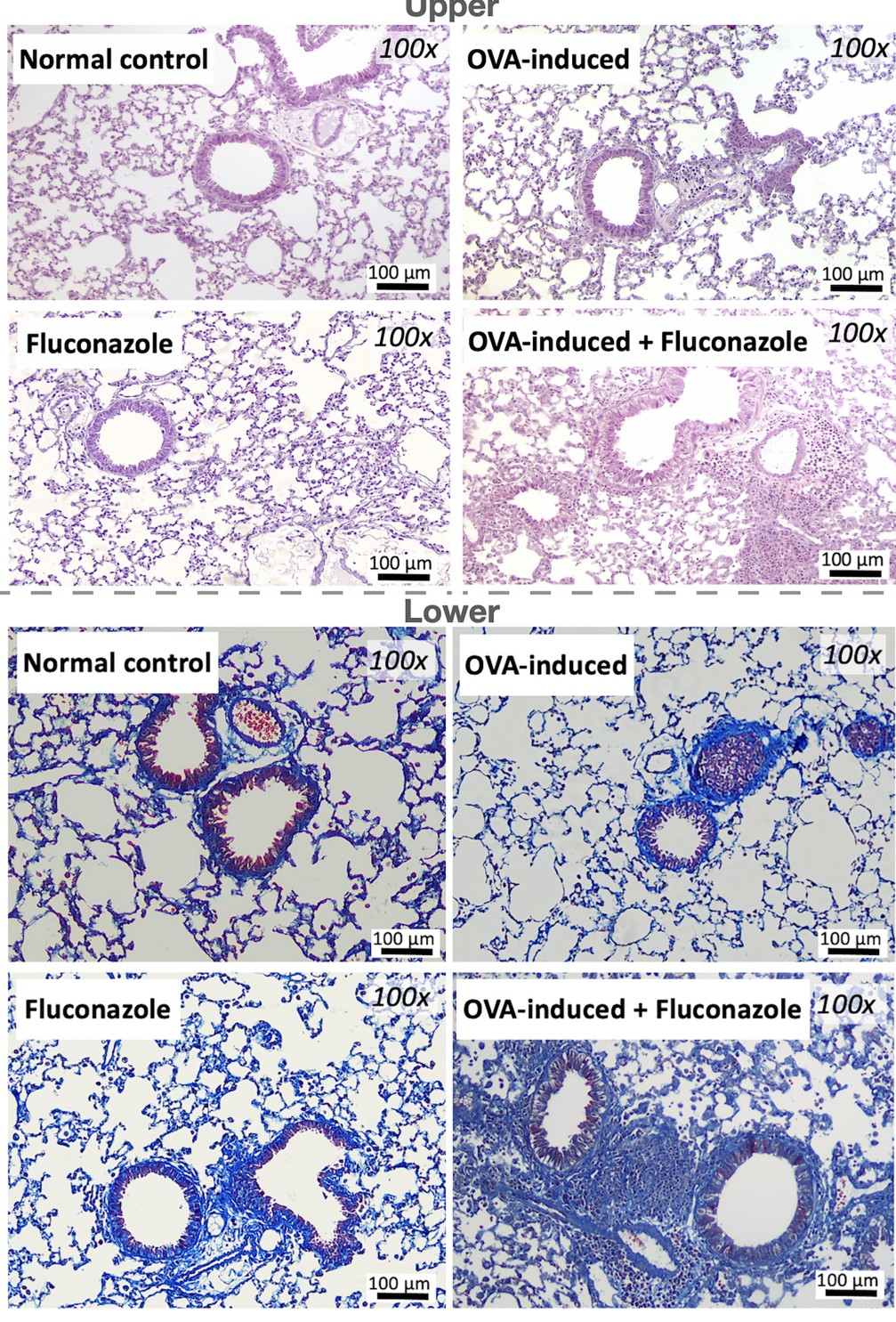

**Figure 2** Representative lung histological pictures of control, asthma, fluconazole in normal mice, and fluconazole in asthma as stained by Hematoxylin and Eosin color (upper part) and Masson's Trichrome (lower part) are demonstrated, ($n$ = 6/group).

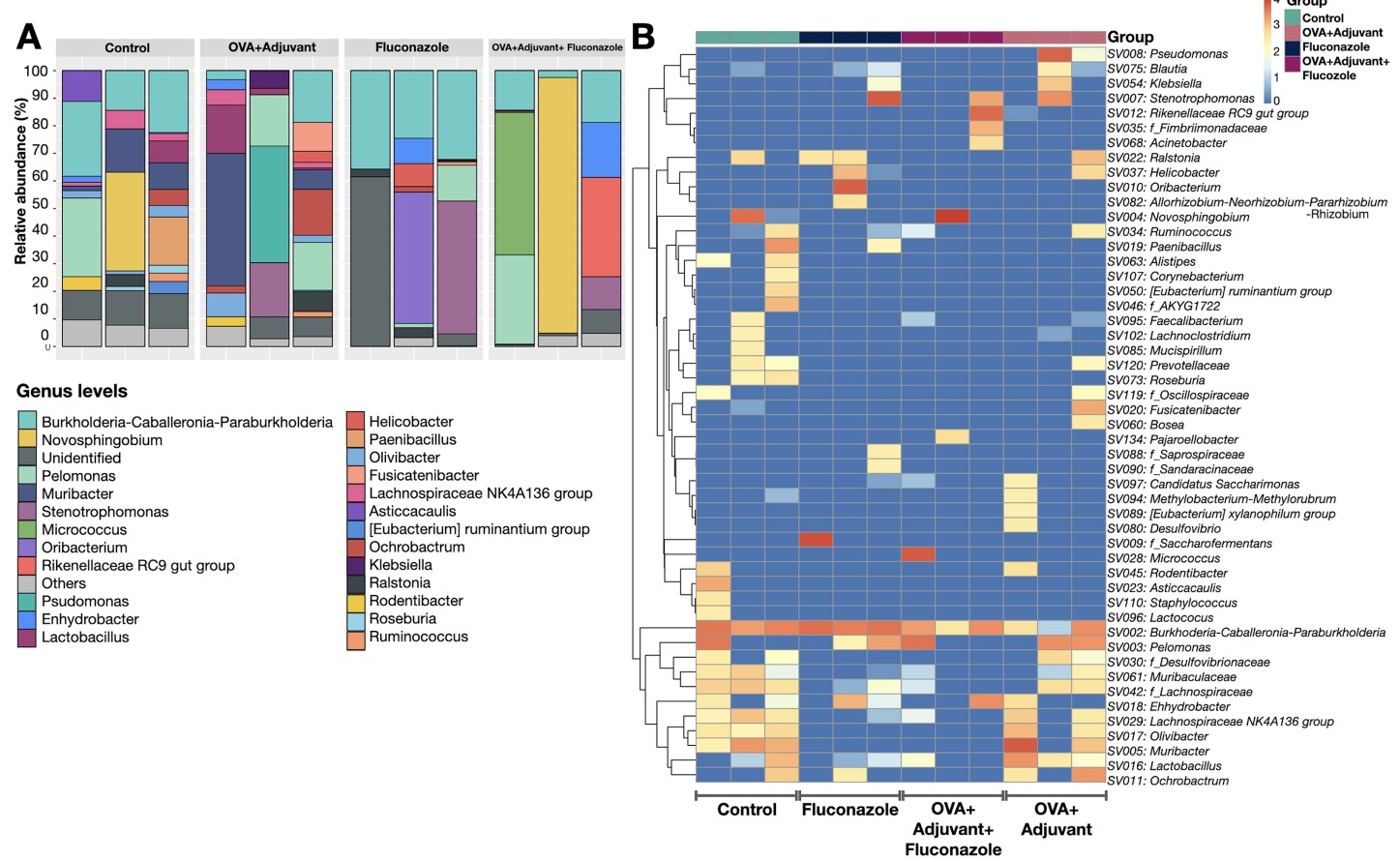

**Figure 3** Microbiome analysis from the lung of control, asthma, fluconazole in normal mice, and fluconazole in asthma as indicated by bacterial abundance in the genus (graph and heat map presentation) (A, B), (*n* = 3/group).

asthma and asthma alone exhibited similar parameters, including lung inflammatory cytokines, mononuclear cells in bronchoalveolar lavage fluid (BALF), and lung damage scores (Figs. 1J–1L, 1N, 1O). Notably, an increase of immune cells was observed in the lungs of asthma mice (Fig. 1N), which may contribute to the elevated levels of TNF-α and IL-6 in the lung tissue. These findings support the characterization of asthma as a local inflammatory disease (*Lemanske, 2000*).

## Lung microbiota alteration in asthma and impacts of fluconazole

Because the local inflammatory responses in OVA-administered mice might partly be due to the alteration of lung microbiota from the induction of asthmatic responses, microbiome analysis from the lung tissue (the lower respiratory tract) was performed, as such, the relative abundance of bacteria in different levels, including the class, order, and family (Fig. S1), and the abundance in the genus (graph and heat map presentations) (Figs. 3A, 3B) together with the beta diversity, using Bray-Curtis distance and Generalized UniFrac (GUniFrac) with an alpha of 0.5 (similarity separation) (Fig. 4A), and the determination of representative bacteria in each group (the linear discriminant analysis; LDA) with the alpha diversity (Figs. 4B, 4C) were performed. For bacterial abundance,

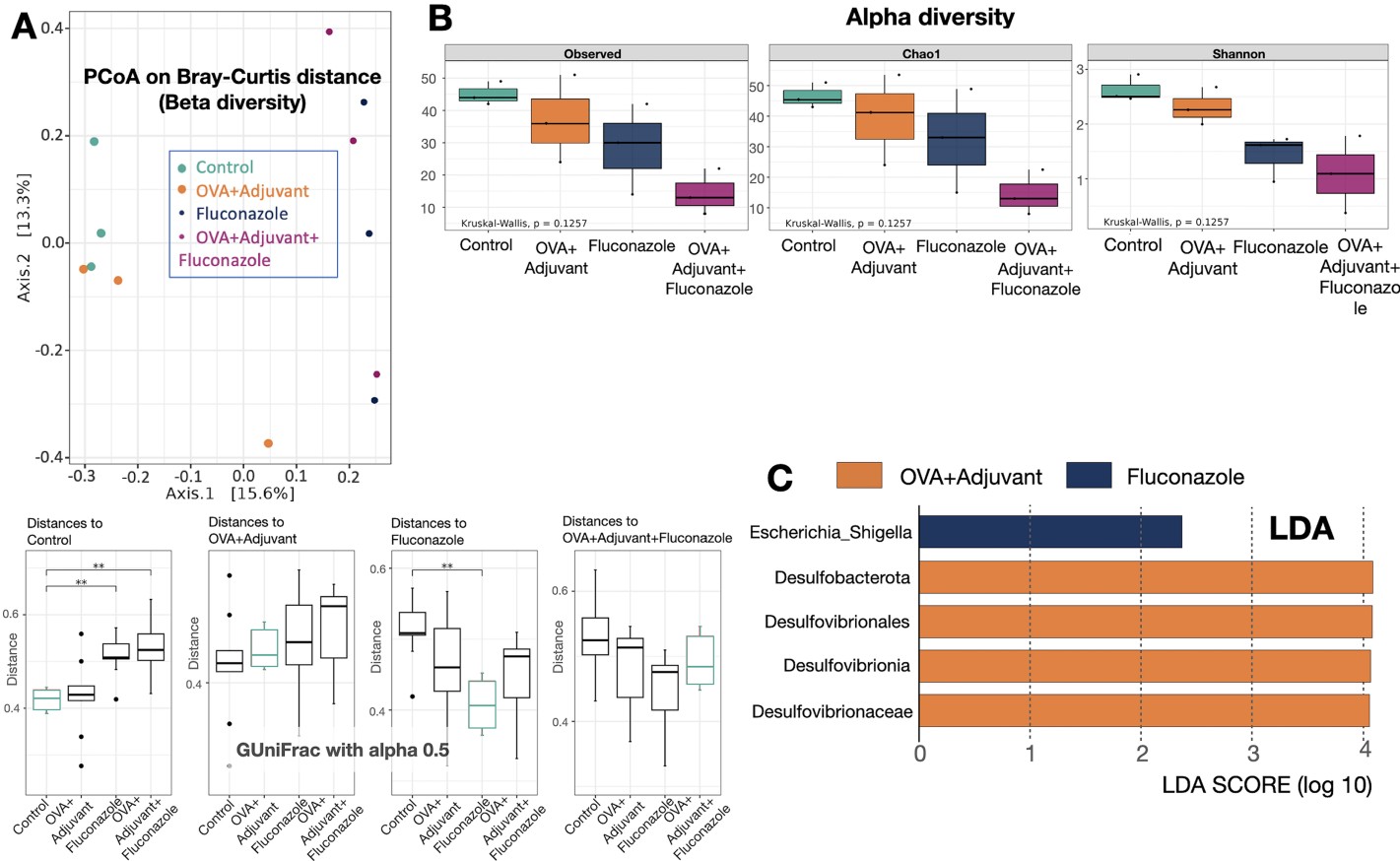

**Figure 4 Microbiome analysis from the lung of control, asthma, fluconazole in normal mice, and fluconazole in asthma as indicated by beta diversity using the principal coordinate analysis (PCoA) based on Bray Curtis dissimilarity (A upper part) with the distance comparisons (A lower part), the alpha diversity, including observed Operational taxonomic unit (OTUs), Chao-1, and Shannon (B), and the linear discriminant analysis (LDA) (C), are demonstrated, ($n$ = 3/group).** The data are the mean ± SE, **; $p < 0.05$ *vs.* Control using ANOVA with Tukey's analysis.

Proteobacteria (mostly Gram-negative aerobic pathogenic bacteria) was dominant in all groups (non-different among groups), followed by Firmicutes (mostly Gram-positive anaerobic beneficial bacteria) (Fig. S1, phylum level).

In fluconazole-treated asthma, there appears to be a lower abundance of Firmicutes than in non-treated asthma and a higher abundance of Bacteroidota (former name was Bacteroidetes; mostly Gram-negative anaerobic bacteria) when compared with fluconazole-treated control (Figs. 5A, 5B and Fig. S1, phylum level). The increased Firmicutes in the Fluconazole group compared to the control, along with the reduced fungi (Fig. 1M), support the impacts of fungi on bacterial population (*Medeiros et al., 2023*). Surprisingly, there seems to be no difference in the abundance of Proteobacteria in the lungs between the control and asthma groups (Fig. 5C). The increased Bacteroidota with decreased Firmicutes in fluconazole-asthma (Fig. 5B) might elevate immune responses against Bacteroidota (Gram-negative bacteria) because of the potent innate immune activation of LPS (the major component of Gram-negative bacterial cell wall) (*Alexander & Rietschel, 2001*), that possibly worsen the severity of asthma. However, the

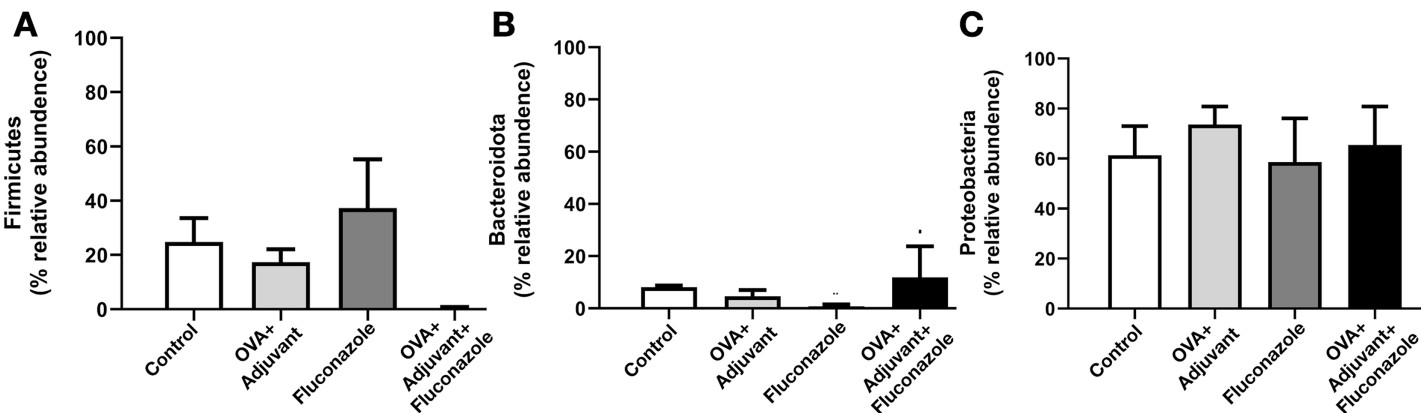

**Figure 5** (A–C) Selected bacterial abundance from the lung microbiome analysis of control, asthma, fluconazole in normal mice, and fluconazole in asthma with the graph presentation (from Figs. S1 and 3A) are demonstrated, (*n* = 3/group).

LDA score indicated *E. coli*-Shigella as the representative bacteria in mice with fluconazole alone, and several Desulfobacteria represented the asthma control group without the representative bacteria in other groups (Fig. 4C).

Although the bacterial abundance among groups was only subtly different, the beta and alpha diversities (Figs. 4A and 4B) indicated notable differences. For beta diversity (an alteration in bacterial species), the principal coordinate analysis (PCoA) based on Bray Curtis dissimilarity metrics (a simplified method to demonstrate the similarity among groups using the 2-dimensional distances from the axis) indicated the similarity among normal control mice (green dots) (Fig. 4A, upper part). There was a clear difference between i) control and fluconazole alone and ii) control and fluconazole-asthma but not between control and asthma alone, as calculated by the GUniFrac with alpha 0.5 (Fig. 4A, lower part; distance to control and distance to fluconazole), showed that microbiota communities were separated among the sample groups (PERMANOVA test; *p* = 0.03796). However, other beta-diversity plots showed no significant difference among the groups. In parallel, the alpha diversity (the species diversity or richness of the species within the community), using observed operational taxonomic unit (OTUs), Chao-1, and Shannon score, indicated the highest bacterial diversity in the control group, while fluconazole-asthma indicated the lowest diversity (Fig. 4B). There was no statistically significant difference among the sample groups (Kruskal Wallis test; *p* > 0.05) (Fig. 5B). These data indicated a subtle change in lung microbiome between control and asthma but a more prominent alteration with the use of fluconazole (Figs. 4A–4C) with the direction toward reduced Firmicutes and increased Bacteroides (Figs. 5A, 5B).

## The pro-inflammatory impact on lung cells with the presence of Lipopolysaccharide and β-glucan

Due to the elevated fungi from asthma (Fig. 1M), perhaps through the transfer of oropharyngeal *Candida* into the lung during respiratory distress (*Gani et al., 2020*), there seems to be an increase in Bacteroides in the fluconazole-asthma group (Fig. 5B), the

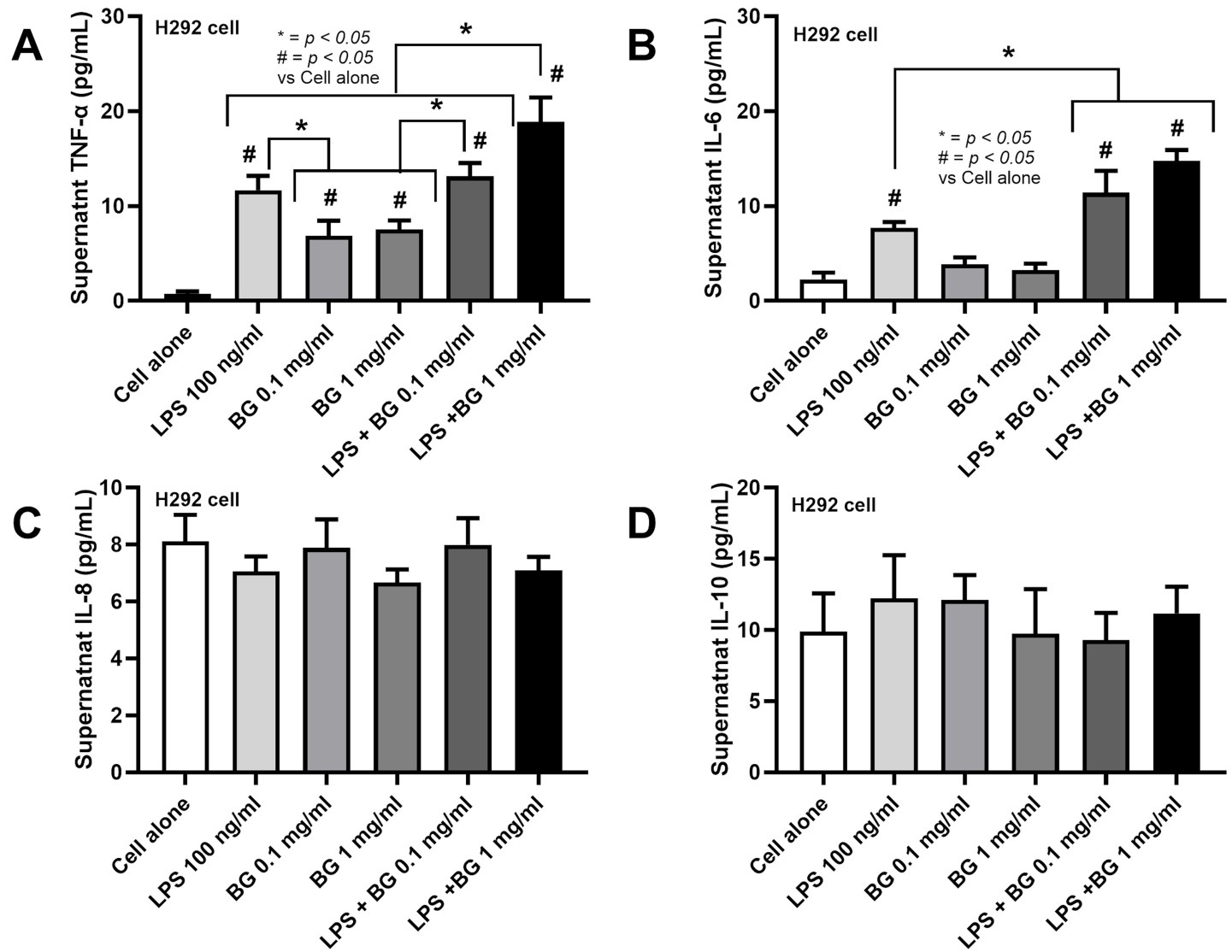

**Figure 6 Characteristics of pulmonary cells (H292) after the activation by lipopolysaccharide (LPS) with or without beta-glucan (BG) in low (0.1 mg/mL) and high (1 mg/mL) doses as indicated by supernatant cytokines (TNF-α, IL-6, IL-8, and IL-10) are demonstrated (data were derived from triplicated independent isolated experiments), (n = 3/groups with three independent repeats).** The data are shown as the mean ± SE, #; $p < 0.05$ vs. Cell alone, and $^{*}$; $p < 0.05$ using ANOVA with Tukey's analysis.

co-presentation of LPS and β-glucan (BG), the major cell wall components of Gram-negative bacteria and fungi, respectively, are possible. In comparison with LPS alone, the additive pro-inflammatory effect of LPS and BG (1 mg/mL) on H292 cells (the pulmonary cell line) was demonstrated by TNF-α and IL-6 but not on IL-8 and IL-10 (Figs. 6A–6D). The highest TNF-α production was shown in LPS plus high dose BG (1 mg/mL) (Fig. 6A), while BG alone activated only TNF-α but not IL-6 (Figs. 6A, 6B). All stimulations had no response with IL-8 and IL-10 (Figs. 6C, 6D). These data suggest a potential additive pro-inflammatory effect of fungi and bacteria, potentially leading to heightened pulmonary cell inflammation and the worsening of severe asthma.

## DISCUSSION

Although antifungal drugs, including fluconazole, are useful for the treatment of allergic bronchopulmonary aspergillosis (ABPA), also known as fungal asthma or airway mycosis or severe asthma with fungal sensitization (SAFS) (*Agarwal & Gupta, 2011*), and the increased abundance of *Candida* spp., especially in saliva, in patients with chronic asthma is mentioned (*Abidullah et al., 2022*), the use of fluconazole might worsen lung inflammation in non-fungal-correlated asthma, partly through the alteration of bacterial microbiota in the lung.

Here, there was an upregulation of fungal genes in the lung of OVA-induced asthma mice with PCR using ITS primers; however, without positive fungi in the culture method, the abundance of fungi in this model possibly was too low to be detectable by culture. Due to the technical limitation of ITS primers, types of fungi could not be identified. Nevertheless, numerous airborne fungi are implicated, including species such as *Alternaria* spp., *Aspergillus* spp., *Cladosporium* spp., *Penicillium* spp., and *Candida* spp. (*Denning et al., 2006*), which are received from the environments and endogenous sources (gastrointestinal tract), respectively. In our mice, *Candida* spp. from the intestines is the most likely possible because of the environmental protection of the animal facility. Due to the subtle elevated fungi in OVA-induced asthma, the bacterial population in IgE-mediated asthma mice was similar to the control group, with only a trend of lower bacterial diversity (alpha diversity). Fluconazole treatment in control mice induced *E. coli*-Shigella (LDA score), and the difference between fluconazole in normal and fluconazole-asthma was indicated only by increased Bacteroidota and lower Shannon score (alpha diversity) in fluconazole-asthma. Although the data generated by NGS sequencing should not be compared directly using standard statistical methods, this study suggests that fluconazole may have eliminated fungi with detrimental effects on Bacteroidota. Due to the limitation of the sample size, future studies should be investigated. The elevation of LPS from Bacteroidota (Gram-negative bacteria) and increased BG from the dead fungi (fluconazole fungicidal activity) might synergistically activate innate immune responses as indicated in pulmonary cells, here and in neutrophils and macrophages in other published articles (*Saithong et al., 2022*). Indeed, the additive impact of LPS plus BG compared with LPS alone is previously mentioned partly through the crosstalk between Toll-like receptor 4 (TLR-4) and Dectin-1, the main receptors for LPS and BG, respectively (*Onyishi et al., 2023*). Although BG alone is not a potent immune activator, BG synergistically enhances the action of LPS, a powerful innate immune stimulator, as previously indicated by the increased cell energy responses of innate immune cells (neutrophils and macrophages) (*Issara-Amphorn et al., 2021*). Accordingly, the evidence of increased inflammatory responses in fluconazole-treated asthma mice, partly due to the co-presentation of LPS and BG, was demonstrated through lung pathology, lung cytokines, and prominent mononuclear cells in fluconazole asthma over asthma alone. The elevated fungal levels alone, without a corresponding increase in Bacteroidota, in the asthma-only group may elicit a weaker immune response compared to the fluconazole-asthma group, which shows elevated levels of both factors. This difference

may be attributed to the less potent immune-stimulating properties of beta-glucan (BG) (Fig. 6).

From a clinical perspective, the use of fluconazole in asthma should be limited only to patients with fungal-related asthma in several classifications. Or regular asthma with positive IgG aspergillus precipitin due to common exposure to *Aspergillus* spp. from the environment (*Faux, Shale & Lane, 1992*). The unnecessary use of fluconazole for asthma or the treatment of non-pulmonary fungal infection in patients with asthma might need more careful monitoring due to the possibility of worsening inflammation and asthma severity. Additionally, elevated asthma severity might be attributed to the accumulation of certain bacteria in the lungs. Consequently, the attenuation of lung dysbiosis and the gut-lung microbiome axis could be topics for future exploration.

## CONCLUSIONS

This study demonstrated increased fungal presence in OVA-induced asthma mice and enhanced lung inflammation with fluconazole treatment, partly due to fluconazole-induced lung dysbiosis. The use of fluconazole in patients with asthma may require closer monitoring of lung function in the future.

## ACKNOWLEDGEMENTS

We extend our gratitude to Ariya Chindamporn for her exceptional advice and technical assistance.

### Funding

This research was supported by the Research and Diagnostic Center for Emerging Infectious Diseases (RCEID), Faculty of Medicine, Khon Kaen University and Mycology laboratory, Department of Microbiology, Faculty of Medicine, Chulalongkorn University. The funders had no role in study design, data collection and analysis, decision to publish, or preparation of the manuscript.

### Grant Disclosures

The following grant information was disclosed by the authors:
Research and Diagnostic Center for Emerging Infectious Diseases (RCEID).
Faculty of Medicine, Khon Kaen University and Mycology laboratory.
Department of microbiology, Faculty of Medicine, Chulalongkorn University.

### Competing Interests

The authors declare that they have no competing interests.

### Author Contributions

- Jesadakorn Worasilchai conceived and designed the experiments, performed the experiments, prepared figures and/or tables, authored or reviewed drafts of the article, chemical preparations, and approved the final draft.

- Piyapat Thongchaichayakon conceived and designed the experiments, performed the experiments, prepared figures and/or tables, authored or reviewed drafts of the article, and approved the final draft.
- Kittipat Chansri conceived and designed the experiments, performed the experiments, prepared figures and/or tables, authored or reviewed drafts of the article, and approved the final draft.
- Supichaya Leelahavanichkul analyzed the data, authored or reviewed drafts of the article, and approved the final draft.
- Vathin Chiewvit analyzed the data, prepared figures and/or tables, authored or reviewed drafts of the article, and approved the final draft.
- Peerapat Visitchanakun performed the experiments, analyzed the data, prepared figures and/or tables, authored or reviewed drafts of the article, and approved the final draft.
- Poorichaya Somparn analyzed the data, authored or reviewed drafts of the article, and approved the final draft.
- Pratsanee Hiengrach conceived and designed the experiments, performed the experiments, analyzed the data, prepared figures and/or tables, authored or reviewed drafts of the article, chemical preparations, and approved the final draft.

### Ethics

The following information was supplied relating to ethical approvals (*i.e.*, approving body and any reference numbers):

Institutional Animal Care and Use Committee of the Faculty of Medicine, Chulalongkorn University, under NIH guidelines

### Data Availability

The mouse lung 16S rRNA sequences are available at NCBI: PRJNA1019875.

### Supplemental Information

Supplemental information for this article can be found online at http://dx.doi.org/10.7717/peerj.18421#supplemental-information.

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
