# Peer review of "Fluconazole worsened lung inflammation, partly through lung microbiome dysbiosis in mice with ovalbumin-induced asthma"

_PeerJ, doi:10.7717/peerj.18421_

## Round 0.1 · original submission · Major Revisions

Two expert reviewers have indicated their concerns with the study and provide good suggestions for a revision.

Reviewer 1 ·

Basic reporting

The document is written in English but could use some editing to clarify some of the sentences. I suggest using a free tool such as Grammarly to improve upon sentence structure.

The manuscript uses adequate references. However, on line 48, the authors describe asthma as a T2 endotype. More appropriately, the authors should clarify that this type of asthma is usually classified as allergic asthma and that other endotypes do exist.

On lines 57-59, the sentence structure is somewhat misleading in suggesting that the eosinophils are a part of the adaptive immune response. I understand that the authors were trying to highlight the T lymphocyte population and their role in the eosinophilic response, but the sentence structure should be revisited for clarity.

The only time the authors discuss severe asthma with fungal sensitization (SAFS) is in the discussion. However, it should be discussed earlier as a part of the introduction when introducing fluconazole's use in the asthma field (i.e. lines 77-80).

While Aspergillus fumigatus is the main causative agent of SAFS, it is not the only cause. Therefore, the statement on lines 245-246 is misleading.

The hypothesis is not clearly stated.

Several of the histological images are blurry and the OVA-induced and fluconazole Trichrome images look like they have been white balanced too much.

Experimental design

The study does not list a hypothesis. All that is stated is a brief statement that describes the current knowledge gaps in the current literature.

There is no justification as to why the authors chose to only use male mice.

Was the environment used for housing the mice specific pathogen free?

What was the standard bedding (wood, corn, paper), standard food (source, autoclaved or not), and was the water autoclaved? All of these conditions can alter the microbiome and as such should be listed.

It is not clear regarding the n used for this study. As well, it is not stated whether the studies were repeated.

What concentration of fluconazole?

Validity of the findings

I cannot conclude the validity of the findings based on what is currently provided within the manuscript. Without knowing the "n" for the study it is difficult to assess the accuracy of the statistical tools used. I think this is especially important as the authors mention in bold statements about the clinical use of fluconazole in asthmatic patients within the paragraph starting on line 279.

Also, I am unsure of the meaning behind the final sentence of the discussion (line 287) "More studies are interesting.".

Reviewer 2 ·

Basic reporting

No comment

Experimental design

No comment

Validity of the findings

The manuscript by Worasilchai et al. deals with the effects of the antifungal fluconazole in a murine model of OVA-induced asthma. The rationale for this study comes from the increasing appreciation of the role of fungi in asthma. To address this issue, mice were characterized for cytokine profile, lung histopathology, fungal content and bacterial communities. In addition, a pulmonary cell line was used for evaluation of cytokine production upon co-stimulation with LPS and β-glucan. The authors conclude that fluconazole may be responsible for the worsening of lung inflammation and advocate its use only for asthma of fungal etiology. In my opinion, the results do not fully support the conclusions. In particular, I have the following concerns that the authors may want to address:
- the experimental setup is not entirely clear to me. In particular, lines 104-105 are confusing. It seems that mice not treated with fluconazole (“Control” and “Asthma” groups) were euthanized on day 23 while fluconazole-treated mice (“Fluconazole control” and “Asthma + Fluconazole”) were euthanized on day 30. I think that all mice were euthanized on day 30, as reported in Figure 1A, otherwise this would introduce a temporal difference between fluconazole-untreated and treated mice that may be relevant for the experimental outcomes.
-If I understand correctly, 6 mice per group were used (line 95). However, metagenomics data are reported for only 3 mice per group. In addition, from the raw data, it seems that weight change, cytokines, IgE, lung ITS and histology were measured from 4 mice. Could the authors please clarify this issue? How many mice per group were used and how selection for analysis was performed?
- line 188: serum IL-6 (Fig. 1F) is increased in asthma mice as also reported in line 172. This sentence should be reworded.
- Figure 5 is critical in my opinion. Since data generated by NGS sequencing are intrinsically compositional, proportions between samples or groups should not be compared directly with standard statistical methods
- lines 217-218: although I substantially agree with this sentence, it should be noted from the raw data that: “The GUniFrac with alpha 0.5 showed that microbiota communities were clearly separated among the sample groups (PERMANOVA test; p = 0.03796). However, other beta-diversity plots were no significant difference among the groups.” and, in terms of alpha diversity “there were no statistically significant difference among the sample groups (Kruskal Wallis test; p>0.05)”. I think that this should be mentioned in the text.
- last paragraph 234-246: i) the title of the section should be changed since no bacteria and fungi were used, but LPS and β-glucan as surrogates; ii) please, clarify among “additive” (line 240) and “synergistic” (line 245): what is the suggested effect of co-stimulation?

Additional comments

Minor comments:
- Excessive use of self-citations throughout the manuscript
- line 124: “previous protocol ()”; please, add reference
- figure 3: taxa should not be ranked by alphabetical order, but in terms of their relative abundance
- figure 3: representation of taxonomy at all levels is unnecessary, also considering that discussion is limited to phyla and genera. Either move the other levels in a Supplementary Figure or consider removing
- figure 4: the overall quality of the panels is low and some graphs are very difficult to read

---

## Round 0.2 · Major Revisions

Whereas I appreciate that the authors have met several points of the reviewers, I concur with reviewer 1 that the sample size still remains a problem.

Reviewer 1 ·

Basic reporting

The information provided is more robust. The authors now state a hypothesis.

Experimental design

My biggest concern is still the sample size. While the authors now state clearly that the sample size is 6 mice/group there is no mention in the text as to the number of independent repeats to ensure data accuracy. In their rebuttal, they mention "Although the studies
were repeated for 2 times to complete all 24 mice, the sample size for each group of mice was also
determined for statistical significance using the G*Power 3. 1 software prior to conducting the
experiments." There is no mention of how they split the mice for the repeated studies (i.e. did they do all of the groups at once with 3/group, or did they split when they completed the study for the different groups?).

Regarding the concentration of fluconazole, the only value they provide is volume. This is too vague and requires a mg/mL concentration to provide readers the information needed to replicate.

Validity of the findings

While these findings are definitely valuable to the field. I am hesitant to accept the immunological findings as I am unsure how the authors conducted their two repeated studies, and I am unsure based on the text written in the manuscript as to the validity of the data as it appears that the immunological was only done once with no independent studies to affirm the conclusions.

Reviewer 2 ·

Basic reporting

No comment

Experimental design

No comment

Validity of the findings

I thank the authors for addressing most of my comments. I have only a few remaining comments that the authors may want to address:
- the authors have removed statistical analysis from the graphs in Figure 5. I understand that the authors support the hypothesis that Fluconazole reduces the relative abundance of Firmicutes while increasing the one of Bacteroidota in asthma (Fig. 5A-B), but I do not understand the rationale behind the selection of the other taxa (Fig. 5D-I). Most of them are not discussed in the text. I would suggest to remove these taxa.
- It is not clear to me why an arrow links the upper and the lower panels of Figure 4A. I would suggest to either use the PCoA on GUniFrac with alpha 0.5 or remove the arrow. If PCoA on Bray-Curtis is kept, please check for a reporting error. It appears that two dots (red and orange) are present close to Axis 2 instead of one in the corresponding position of the source data.

Please, check whether “nucleotide” in line 212 and “Fig.5D” in line 218 are correct.

---

## Round 0.3 · accepted · Accept

Thank you for addressing the reviewers comments the second time around, congratulations

Reviewer 1 ·

Basic reporting

All areas meet the standards of PeerJ.

Experimental design

Clarity is provided regarding the sample size of animals, independent studies, and fluconazole concentration. While the sample size is small for immunologic work, there is enough detail for others to replicate if desired, which I believe is sufficient for publication.

Validity of the findings

No comment

Additional comments

No comment

Reviewer 2 ·

Basic reporting

No comment

Experimental design

No comment

Validity of the findings

No comment